# The Sensitization of Triple-Negative Breast Cancers to Poly ADP Ribose Polymerase Inhibition Independent of BRCA1/2 Mutation Status by Chemically Modified microRNA-489

**DOI:** 10.3390/cells13010049

**Published:** 2023-12-26

**Authors:** Young Hwa Soung, Jingfang Ju, Jun Chung

**Affiliations:** Department of Pathology, Stony Brook Medicine, Stony Brook, New York, NY 11794, USA; younghwa.song@stonybrookmedicine.edu (Y.H.S.); jingfang.ju@stonybrookmedicine.edu (J.J.)

**Keywords:** chemical modification of miR-489 (CMM489), PARP inhibitor, BRCA, homologous recombination (HR), triple-negative breast cancer (TNBC)

## Abstract

Chemoresistance and inefficient therapeutic efficacies in triple-negative breast cancers (TNBCs) are among the major clinical problems in breast cancers. A potential new method to sensitize these tumors to current treatment options is, therefore, urgent and necessary. Our previous studies demonstrated that miR-489 serves as one of the top tumor-suppressing miRs and features downregulated expression in metastatic TNBCs and that the restoration of miR-489 expression in TNBCs effectively inhibits the metastatic potentials of TNBCs both in vitro and in vivo. The chemical modification of miR-489 (CMM489) through the replacement of uracil with 5-FU further enhances the therapeutic potential of miR-489. In the present study, we tested the effects of CMM489 in synergizing DNA damage response (DDR) inhibitors such as PARP inhibitors. CMM489 is particularly effective in sensitizing TNBC cell lines with inherent resistance to PARP inhibitors regardless of BRCA mutation status. One of the anti-cancer mechanisms through which CMM489 synergizes with PARP inhibitors is the blockade of homologous recombination (HR) in TNBC cells upon DNA damage. The results of this study highlight the potential use of CMM489 in combination treatments with PARP inhibitors in TNBCs.

## 1. Introduction

Triple-negative breast cancers (TNBCs) are the most aggressive subtypes of breast cancers and occur more frequently in younger women, with a propensity to metastasize to bone and soft tissues, including the lung and brain [1,2,3,4]. Unlike other forms of breast cancers, there are currently no effective targeted therapies for TNBCs due to the lack of an estrogen receptor (ER) or progesterone receptor (PR) and amplification of the HER2 receptor [5,6]. Thus, polychemotherapy including DNA-damaging agents remains the treatment of choice for TNBCs [5,6]. Identifying TNBC-specific targets is urgently needed to achieve effective therapeutic approaches and overcome therapeutic resistance. As a subtype, TNBCs are enriched in tumors with homologous recombination (HR) deficiencies and depend on an alternative DNA repair system to respond to genotoxic stress [2,7,8]. For this reason, targeting HR defects through synthetic lethality (the impairment of more than two HR components, resulting in cell death) has emerged as a promising strategy through the development of HR inhibitors in cancer treatment, including TNBCs [8].

In particular, TNBCs with germline BRCA1/2 mutations show better sensitivity to HR inhibitors such as the inhibitors of poly ADP ribose polymerase (PARP inhibitors) (i.e., Olaparib), which were FDA-approved in 2018 for the treatment of BRCA germline mutant TNBCs [9,10,11,12,13,14,15,16]. However, TNBCs with a germline BRCA mutant background represent only 15% of the total TNBC population [17]. Further, fewer than 50% of TNBC patients with BRCA mutations respond to PARP inhibitors [18,19]. In addition, TNBCs with functional BRCA genes (85% of TNBCs) do not generally respond well to polychemotherapies and PARP inhibitor treatment [20,21,22]. There are currently four PARP inhibitors (Olaparib, Rucaparib, Niraparib, and Talazoparib) approved by the FDA for the treatment of advanced ovarian and breast cancers with BRCA1/2 deficiency [23,24], and three PARP inhibitors (Veliparib, Pamiparib, and Fluzoparib) are currently under clinical trials for evaluation [25]. Among the four FDA-approved PARP inhibitors, Olaparib and Talazoparib are currently being used for the treatment of breast cancer patients with BRCA1/2 deficiencies [23,25]. Recent studies have demonstrated that a significant portion of TNBCs still show resistance to HR inhibitors such as PARP inhibitors, regardless of their BRCA mutation status [18,19,20,21,22]. Therefore, there is an urgent need to develop novel therapeutic agents to target BRCA-independent HR pathways and sensitize resistant TNBCs to HR inhibitors. Given the high tolerability of PARP inhibitors as single agents [20,21,22], enhancing the use of PARP inhibitors through smart combinations in TNBCs regardless of BRCA status is desirable.

Cell-line-based miR-seq profiling studies led to the identification of miR-489 as a tumor-suppressing miR whose expression is most strongly downregulated in metastatic TNBC cell lines in comparison to normal cell controls and nonmetastatic breast cancer cells [26]. Subsequent studies have demonstrated the clinical relevance of miR-489 by showing that miR-489 expression is downregulated in drug-resistant and metastatic breast cancer tissues and is correlated with patient survival [27]. Our analyses and those of others provide evidence that the restoration of miR-489 expression in metastatic TNBC cells and tumors effectively reduces their metastatic potential [28,29]. Our analyses also demonstrated that the blockade of DNA damage responses in TNBC cells is one of the key anti-cancer mechanisms of miR-489 [29]. To further exploit the therapeutic potentials of miR-489, we chemically modified miR-489 by replacing uracil with 5-fluorouracil (5-FU) as CMM489 (chemically modified miR-489) [29]. Although 5-FU is no longer a chemotherapeutic agent for treating breast cancers, the incorporation of 5-FU into miR-489 further enhances the therapeutic potentials of miR-489 by combining a DNA-damaging agent (5-FU) and a tumor-suppressing component (miR-489) into a single and novel therapeutic entity [29]. Therefore, the rationale behind the development of CMM489 is that the 5-FU component induces DNA damage, and the miR-489 component blocks DNA damage responses such as HR. In the present study, we provide evidence that a combination of these two components enables CMM489 to induce synthetic lethality in TNBCs at low doses that are not toxic to most cell types. The most encouraging aspect of CMM489 is that it is effectively anti-proliferative in a large number of TNBC cell lines with diverse genetic backgrounds, and its IC_50_ value is significantly lower than the values of currently available breast cancer drugs. The results suggest that CMM489 is a superior candidate as a combination partner in TNBC therapies including PARP inhibitors.

## 2. Materials and Methods

### 2.1. Cell Lines and Reagents

All TNBC cell lines used in this study were purchased from ATCC. MDA-MB-231, MDA-MB-436, and Hs578T breast cancer cells were cultured in a DMEM medium with 10% FBS and 1% penicillin/streptomycin. HCC-1937 and BT549 breast cancer cells were cultured in an RPMI-1640 medium with 10% FBS and 1% penicillin/streptomycin. These cells were grown in humidified incubators at 37 °C in 5% CO_2_. Olaparib (Cat. No. S1060), Rucaparib (Cat. No. S1098), and Talazoparib (Cat. No. S7048), dissolved in dimethyl sulfoxide (DMSO), were purchased from Selleckchem (Houston, TX, USA).

### 2.2. Cell Viability Assay and Drug Synergy Assay

Cells (1800–3000/well) were seeded in 96-well plates with 100 μL media in triplicate. The cells were cultured in the presence of Olaparib or CMM489 alone or in combination with Olaparib and CMM489 at different concentrations for 3–7 days. Cell viability was assessed with an MTT assay using Kit-8 (Dojindo Molecular Technologies, Rockville, MD, USA) according to the manufacturer’s instructions. Absorption at 450 nm was determined using an iMark Microplate Reader (Bio-Rad, Hercules, CA, USA). For drug combination studies, IC_50_ values were first determined, and then combination indexes (CI) were calculated using CompuSyn software 1.0.1. (ComboSyn, INC. Paramus, NJ, USA). These CI values were used to evaluate the synergistic effect of two-drug combinations. Here, a CI of <0.7 indicates synergism, 0.7–0.9 indicates slight synergism, 0.9–1 indicates additive synergism, and >1 indicates antagonism.

### 2.3. Immunofluorescence Staining

Cells placed on cover slips were exposed to Olaparib, CMM489, or the combination treatment. After 3 days, the cells were washed in PBS, fixed in 4% formaldehyde, permeabilized in 0.2% Triton X-100, and blocked in 1% bovine serum albumin (BSA) in PBST. The cells were then incubated with primary antibodies (rabbit anti-RAD51; Abcam, Cat. No. ab133534, mouse anti-γHA2X; Abcam, Cat. No. ab26350) for 2 h at room temperature or overnight at 4 °C and incubated with appropriate fluorophore-conjugated secondary antibodies for 1 h at room temperature. Slides were mounted with Vectashield DAPI (Vector lab). Immunofluorescence was visualized using a Nikon Eclipse Ts2R microscope with a Nikon DSQi2 Digital Camera. All images were analyzed using NIS-Elements software (NIS-Elements advanced research 4.5 version, Nikon, Tokyo, Japan).

### 2.4. Western Blot Analysis

Western blots were performed as described previously [1]. In brief, protein lysates were separated on 4% to 20% gradient SDS PAGE and transferred to PVDF membranes (Bio-Rad). The blots were probed with primary antibodies and further with the respective secondary antibodies conjugated with HRP. Antibodies against RAD51 (Cat. No. ab133534) and γHA2X (Cat. No. ab11175) were purchased from Abcam. The Β-actin (clone C-11) antibody was obtained from Santa Cruz Biotechnology (Dallas, TX, USA). Proteins were detected using a Clarity Western ECL blotting substrate (Bio-Rad), and protein bands were imaged with a ChemiDoc Touch Imaging System (Bio-Rad). The band densities were quantified using ImageJ software.

### 2.5. RNA-Seq Analysis

The total RNA from MDA-MB-231 cells treated with CMM489 or the Olaparib combination was isolated using a Trizol reagent (Invitrogen, Waltham, MA, USA) and RNeasy Mini Kit (Qiagen, Redwood City, CA, USA) according to the manufacturer’s instructions. RNA quantity and quality were assessed using a Nanodrop (Thermo Fisher Scientific Inc., Waltham, MA, USA) and Agilent 2100 Bioanalyzer (Agilent Technologies, Palo Alto, CA, USA). A RIN (RNA Integrity Number) value of more than 9 was used for cDNA library construction. Library preparation and RNA sequencing were performed with Novogene (Novogene Corporation Inc., Sacramento, CA, USA). The cDNA libraries were created using a NEBNext^®^ Ultra™ RNA Library Prep Kit for Illumina^®^ (NEB, Ipswich, MA, USA). The quantification of each library was checked with Qubit 2.0. Then, RNA sequencing was performed on an Illumina Nova seq 6000 platform. Bioinformatic analysis to explain the data, differential analyses, and Venn diagram construction were performed using Novogene. Differential Gene Expression analysis between the control and treatment groups was accomplished using the DESeq2R package (1.20.0) with a Log2 fold change ≥1 or ≤−1 and *p*-value ≤0.05. The resulting *p*-values were adjusted using the Benjamini and Hochberg approach for controlling the false discovery rate.

### 2.6. Organoid Culture and Drug Treatment

Organoids derived from TNBC tissue were obtained from ATCC (Cat. No. PDM-92). Breast cancer organoids were resuspended in growth-factor-reduced Matrigel (BD Biosciences, Cat. No. 356231, Franklin Lakes, NJ, USA). In total, a 40 μL drop of the suspension was placed in the center of a well in a 24-well plate. The plate was flipped and incubated at 37 °C for 25 min to allow the Matrigel to solidify as small domes. Organoid medium was added and changed every 3 days and consisted of AdDF++ (Advanced DMEM/F12 containing 1 × Glutamax and 10 mM HEPES) supplemented with 1 × B-27™ Supplement (Gibco BRL, Grand Island, NY, USA), 5 nM Neuregulin 1 (PeproTech, Rocky Hill, CT, USA), 5 ng/mL FGF 7 (PeproTech), 20 ng/mL FGF 10 (PeproTech), 5 ng/mL EGF (PeproTech), 100 ng/mL Noggin (PeproTech), 500 nM A83-01 (Tocris, Minneapolis, MN, USA), 1.2 μM SB202190 (Tocris), 10 mM Nicotinamide (Sigma-Aldrich, St. Louis, MO, USA), 1.25 mM N-Acetylcysteine (Sigma), and 10% R-Sponidin-1 conditioned media. Y7632 (Rock inhibitor) was removed 2–3 d after initial seeding and organoids were passaged using TrypLE Express (Gibco) every 4 weeks. For drug treatment, 15 μL drops of an organoid/Matrigel suspension were seeded in 96-well white optiplates. When confluency was reached, CMM489, Olaparib, and a combination (Ola + CMM489) containing the medium was added in duplicate for 7 days. To measure cell viability, the medium was removed, and a CellTiter-Glo 3D Reagent (Promega, Fitchburg, WI, USA) was used according to the manufacturer’s instructions. Luminescence was measured in a Spectra MaxM2 plate reader.

### 2.7. siRNA Transfection

MDA-MB-231 cells were depleted of NEK8 using RNAiMax (Invitrogen) according to the instructions of the manufacturer. Custom siRNA sequences targeting NKE8 were synthesized via Dharmacon (NEK8: 5′-TCACTCTTCTGGTTGTAGG-3′). Silencer Negative Control #2 siRNA (control siRNA) was purchased from Ambion (Austin, TX, USA).

### 2.8. Quantitative Real-Time PCR (qRT-PCR) of NEK8

The total RNAs of cells were extracted using a Trizol reagent and Qiagen column purification. In total, 2 μg total RNA was then reverse transcribed into cDNA using a high-capacity cDNA reverse transcription kit according to the manufacturer’s instructions. Real-time PCR was performed using a TaqMan Universal PCR Master Mix with a TaqMan Gene Expression Assay (Applied Biosystems, Waltham, MA, USA) specific to GAPDH (assay ID: Hs02786624_g1) and NEK 8 (assay ID: Hs01093404_m1). mRNA levels were analyzed on an ABI 7500 Real-Time PCR System using the QuantStudio program. NEK8 expression levels were normalized to GAHPD levels. The results were analyzed via the 2^−ΔΔCt^ method. The experiment was performed in triplicate.

### 2.9. Soft Agar Colony Assay

Cells were suspended in the top layer of the DMEM medium containing 0.35% low- melt agarose (ISC Bioexpress, Kaysville, UT, USA) with or without drugs and plated in triplicate on a firm 0.7% agarose base in 6-well plates. The cells were fed twice per week with a DMEM medium containing drugs. Colonies of cells were allowed to grow over the course of 4 weeks. Images of the colonies were obtained using a digital camera mounted to a microscope (Nikon Eclipse Ts2R). The total number of colonies was quantified by counting 50 fields per well using bright-field optics with a grid. The colony formation was performed in triplicate.

## 3. Results

### 3.1. CMM489 Synergizes with Olaparib as a Combination Treatment Partner in TNBC Cells Regardless of the BRCA Mutation Status

Based on our previous study showing the low IC_50_ values of CMM489 in TNBC cells [29], CMM489 has considerable potential to further lower the doses of other drugs as a combination therapeutic agent in breast cancer therapy due to its low toxicity and high efficacy. Therefore, we tested the potential of CMM489 as a combination partner using Olaparib, one of the PARP inhibitors that is an FDA-approved drug for TNBC patients with germline BRCA mutations [18,19,20]. In addition, PARP inhibitors are tolerable agents in drugs used for TNBC treatments [18,19,20]. We used three BRCA wild-type TNBC cell lines and two BRCA mutant TNBC cell lines with various reported sensitivities to Olaparib (Table 1). To assess the synergy, we measured the combination index (CI) at IC_50_ using CompuSyn software. A CI value of 0.9 or lower was considered to indicate synergy, and a CI value higher than 0.9 indicated additive effects. The combination indexes of the CMM489 and Olaparib combination treatment in all TNBC cell lines were all less than 0.9, indicating synergy (Figure 1). Notably, synergy between CMM489 and Olaparib was observed in all three TNBC cell lines with intrinsic resistance to PARP inhibition, such as MDA-MB-231 and BT549 (wild-type BRCA) and HCC1937 (BRCA mutant), suggesting that CMM489 can overcome resistance to PARP inhibitors regardless of BRCA status (Figure 1a,b). The results also suggest that CMM489’s inclusion in PARP inhibitor treatment can dramatically increase CMM489’s viability for the majority of TNBC patients. CMM489 IC_50_ values that showed synergy with Olaparib ranged from 5–70 nM (Figure 1a,b). On the other hand, 50 nM of 5-FU alone did not decrease the viability of MDA-MB-231 cells (Figure 1c), which indicates that the 5-FU component of CMM489 is not toxic when 5-70 nM of CMM489 is used. However, 50 nM of 5-FU alone did not synergize with Olaparib and reduce the viability of these cells. These data suggest that a combination of the 5-FU and miR-489 components (not 5-FU alone) in CMM489 induces synergy with Olaparib and kills TNBC cells, regardless of BRCA mutation status. The data in Figure 1 were acquired from five independent assays as the mean ± SD (n = 5), and CMM489 treatment was performed without transfection agents (vehicle-free).

### 3.2. CMM489 Synergizes with Multiple FDA-Approved PARP Inhibitors in Colony-Forming Assays

Based on the previous report that the effects of PARP inhibitors are more pronounced in long-term survival colony-forming assays than in short-term cell viability assays [30], we tested the effects of CMM489 in combination with multiple FDA-approved PARP inhibitors such as Olaparib, Rucaparib, and Talazoparib in the colony-forming assay (Figure 2). We tested the various doses of CMM489 and each of the PARP inhibitors and found that 10 nM of CMM489, 5 μM of Olaparib and Rucaparib, and 200 nM of Talazoparib offered the most effective synergism. These doses agree with the range shown by the CI values in Figure 1. The synergy between CMM489 and three PARP inhibitors was obvious based on microscopic images of the colonies (Figure 2). The total number of colonies is quantified in Figure 2.

This outcome suggests that the synergistic effects of CMM489 with PARP inhibitors can be observed not only in short-term viability assays but also under long-term survival conditions. We also demonstrated that CMM489 can be combined with various PARP inhibitors to generalize the sensitization effects of CMM489 in TNBCs to PARP inhibition.

### 3.3. Homologous Recombination Deficiency (HRD) Is the Key Mechanism of Synergy between CMM489 and Olaparib

Previous efforts in the development of targeted therapies for TNBCs have been mainly focused on TNBCs with germline BRCA1/2 mutations [20,21,22]. Therefore, other than BRCA deficiencies, the vulnerabilities of HR in TNBCs (especially BRCA wild-type) are not well understood and remain understudied. Our previous analyses demonstrated that blockading DNA-damaging responses is an important anti-cancer mechanism for CMM489 [29], which provides the rationale for studying the relationship between HRD and CMM489. Recent studies using a larger number of TNBC cell lines showed that BRCA deficiencies do not necessarily correlate with sensitivities to PARP inhibition [31]. These outcomes indicate that CMM489 may target BRCA-independent HR pathway(s) to induce HR deficiencies, leading to synthetic lethality. To test this hypothesis, we monitored the inter-nuclear levels of RAD51 and γHA2X in two BRCA wild-type TNBC cell lines (MDA-MB-231 and HS 578T) and two BRCA mutant TNBC cell lines (MDA-MB-436 and HCC1937) upon treatment with Olaparib, CMM489 alone, or a combination of both drugs (Figure 3). Previous reports demonstrated that decreases in RAD51 inter-nuclear levels and increases in γHA2X inter-nuclear levels are associated with deficiencies in HR upon DNA damage [32,33]. As shown in Figure 3, RAD51 disappeared in the nucleus upon treatment with either Olaparib or CMM489. On the other hand, inter-nuclear levels of γHA2X were significantly elevated when Olaparib and CMM489 were treated in combination in all four TNBC cell lines tested (Figure 3a). The outcome was also confirmed via Western blotting analysis (Figure 3b). These results correlate well with the synergistic patterns of these two compounds in these two TNBC cell lines (Figure 1) and provide the rationale for combining these two drugs to maximize the inhibition of HR in TNBCs.

### 3.4. Synergistic Effects of CMM489 and Olaparib Were Observed in TNBC Organoid Models

Patient-derived organoids have emerged as an accurate and effective system for drug screening to select the most effective treatment regimens for patients [34,35]. To test the synergy efficacy of CMM489 and Olaparib in a more clinically relevant model, organoids derived from TNBC tissue were obtained from ATCC (Cat. No. PDM-92) and successfully cultured (Figure 4a). We observed a reduction in TNBC organoids by CMM489 and Olaparib alone (Figure 4b). A significant reduction in the viability of TNBC organoids was observed when CMM489 and Olaparib were treated together (Figure 4b). The results confirmed the synergistic effects of CMM489 and Olaparib in the TNBC organoid model.

### 3.5. RNA-Seq Analysis Identifies NEK8 as a Potential CMM489 Target Gene That Mediates Homologous Recombination

To systematically analyze cellular transcriptome alterations induced by CMM489 or CMM489/Olaparib combination treatment, we used RNA-seq analysis of the total RNA from MDA-MB-231 cells treated with CMM489 alone or CMM489 and Olaparib in combination. The cDNA library preparation and RNA sequencing were performed as described in the Materials and Methods section. The hierarchical clustering heat map shows an overview of differentially expressed patterns among the control, CMM489 treatment group, and CMM489/Olaparib cotreatment group (Figure 5a). Volcano plots showed that 730 genes were specifically upregulated, and 878 genes were significantly downregulated, in MDA-MB-231 cells treated with CMM489 (Figure 5b). The co-expressed genes related to the CMM489 and Olaparib combination were identified using Venn diagram software. There were 11,634 overlapping genes found between the control and CMM489 and 11,475 overlapping genes in the control, CMM489, and Olaparib combination groups (Figure 5c). The significantly differentially expressed genes (DEGs) were determined by fold-change filtering (|log_2_(fold change)| > 1) and *p*-value < 0.05. The top 15 genes downregulated upon treatment with CMM489 are presented in Table 2. 

Among the top 15 genes in Table 2, NEK8 stands out for its role in regulating DNA-damage-induced RAD51 foci formation and replication fork protection [36,37]. While the expression of NEK8 is significantly downregulated by CMM489, 3′UTR of NEK8 does not contain the binding site for miR-489, suggesting that NEK8 is not a direct target gene of miR-489. However, the overall change in NEK8 expression by CMM489 indicates that NEK8 is an important downstream mediator of CMM489’s action, although it is not a direct target gene of CMM489.

### 3.6. Knockdown of NEK8 Expression by siRNA Mimics the Action of CMM489

To understand the role of NEK8 in the HR deficiency created by CMM489, we knocked down the expression of NEK8 by siRNA (Figure 6b). We confirmed the reduction in NEK8 expression under CMM489 (Figure 6a) and NEK8 siRNA treatment (Figure 6b) via q-RT-PCR. Knocking down NEK8 expression with siRNA led to a decrease in RAD51 inter-nuclear levels and an increase in γHA2X inter-nuclear levels, suggesting its role in maintaining HR (Figure 6c). γHA2X inter-nuclear levels are significantly more elevated by Olaparib in MDA-MB-231 cells treated with NEK8 siRNA than in cells treated with control siRNA (Figure 6c). The results indicate that a reduction in NEK8 expression sensitizes TNBC cells to PARP inhibitors and mimics the action of CMM489 (Figure 6c). While NEK8 is one of downstream effectors of CMM489 action, it is likely to play an important role in mediating HR proficiency and deserves consideration in therapeutic targeting for HR inhibitors.

## 4. Discussion

The results showing that CMM489 effectively overcomes the inherent resistance of TNBC cells and tumor organoids to PARP inhibition regardless of BRCA status is highly encouraging because CMM489/PARPi combination treatment could provide a solution for TNBC patients who do not have positive outcomes after neoadjuvant chemo-treatment (NATC). Tumors from such patients showed greater resistance to most available breast cancer drugs, and their prognosis is very poor [7,8]. The present study provides the basis for identifying additional novel combination partners with CMM489 in the future, whereas the current focus has been on characterizing CMM489/PARPi combinations.

Notably, CMM489 is a new and unique drug that is a different and independent therapeutic entity from 5-FU. However, 5-FU is not a treatment option in most breast cancer therapies. The focus of our study was to test the effects of CMM489 as a combination treatment partner compared to those of currently available treatment options such as PARP inhibitors, but not 5-FU. Our previous studies showed that the anti-proliferative effects of CMM489 are as low as 1-50 nM, addressing the concern that the 5-FU component of CMM489 is not randomly toxic in most cells at this dosage. Rather, the incorporation of 5-FU in miR-489 creates a novel therapeutic entity that can suppress a number of key targets involved in HR pathways that inhibit chemoresistant TNBC cells and tumors in a highly selective manner without random toxicities.

Although homologous recombination deficiencies (HRDs) were initially described in cancers with germline mutations of BRCA1/2 [9,10,11], the genetic and epigenetic inactivation of other homologous recombination components can also lead to HRDs [12,13]. Therefore, identifying additional HRD-related genes/pathways targeted by CMM489 will provide important insights into the chemoresistance of TNBCs. The FDA approval of PARP inhibitors for the treatment of TNBCs with germline BRCA mutations was based on numerous clinical trials and pre-clinical studies showing that PARP inhibition induces synthetic lethality with a loss of BRCA1/2 functions in BRCA mutant TNBCs. However, a recent study indicated that BRCA status is not correlated with the sensitivity of a given TNBC cell line to PARP inhibitors [23]. Therefore, the investigation of HRD status following CMM489 or CMM489/PARPi treatment in a larger number of TNBC cell lines with diverse genetic backgrounds will provide insights into novel HR pathways that are related to intrinsic and acquired resistance to PARP inhibition in TNBCs.

The concept of synthetic lethality associated with PARP inhibitors comes from the inability of TNBC cells (due to the disabled signaling pathway) to repair the single-stranded breakage of DNA caused by PARP inhibitors. In this regard, the CMM489-mediated inhibition of downstream genes or pathways important for DNA damage repair likely explains the synergistic effects that produce the synthetic lethality of TNBC cells. We mainly focused on the target genes and pathways of CMM489 rather than those of CMM489 and Olaparib combination treatment because combination treatment likely leads to the large-scale expression of cell stress response genes prior to apoptosis. As Olaparib induces single-stranded breakage, and CMM489 blocks the DNA repair process, synergy likely emerges from the CMM489 downstream target gene that mediates DNA-damaging responses such as HR. The screening of downstream genes and pathways that are significantly regulated by CMM489 led to the identification of NEK8 as a potential target gene of CMM489 in regulating homologous recombination. While the roles of NEK8 in the regulation of HR were reported, our studies established NEK8’s role in mediating the chemo-sensitizing effects of CMM489.

Finding vulnerabilities to induce synthetic lethality in chemoresistant TNBCs has been a major barrier in breast cancer biology. Defining the anti-cancer mechanisms of CMM489 will provide novel insights into how to overcome the treatment resistance associated with TNBCs regardless of BRCA mutation status. TNBCs are more prevalent among young African American females and lead to mortalities among the entire breast cancer population. Therefore, CMM489 will help address the health disparity among the population most susceptible to TNBCs [2,4,11,12]. CMM489 could provide a perfect solution in the development of targeted therapies for TNBC patients with inherent or acquired resistance to inhibitors of HR such as PARP inhibitors.

## 5. Conclusions

Chemically modified miR-489 (CMM489) can serve as a homologous recombination (HR) inhibitor either in monotherapy or in combination therapy with PARP inhibitors such as Olaparib in TNBCs. The characterization of CMM489 as a combination treatment partner with PARP inhibitors in TNBCs meaningfully advances TNBC therapeutics.

## Figures and Tables

**Figure 1 cells-13-00049-f001:**
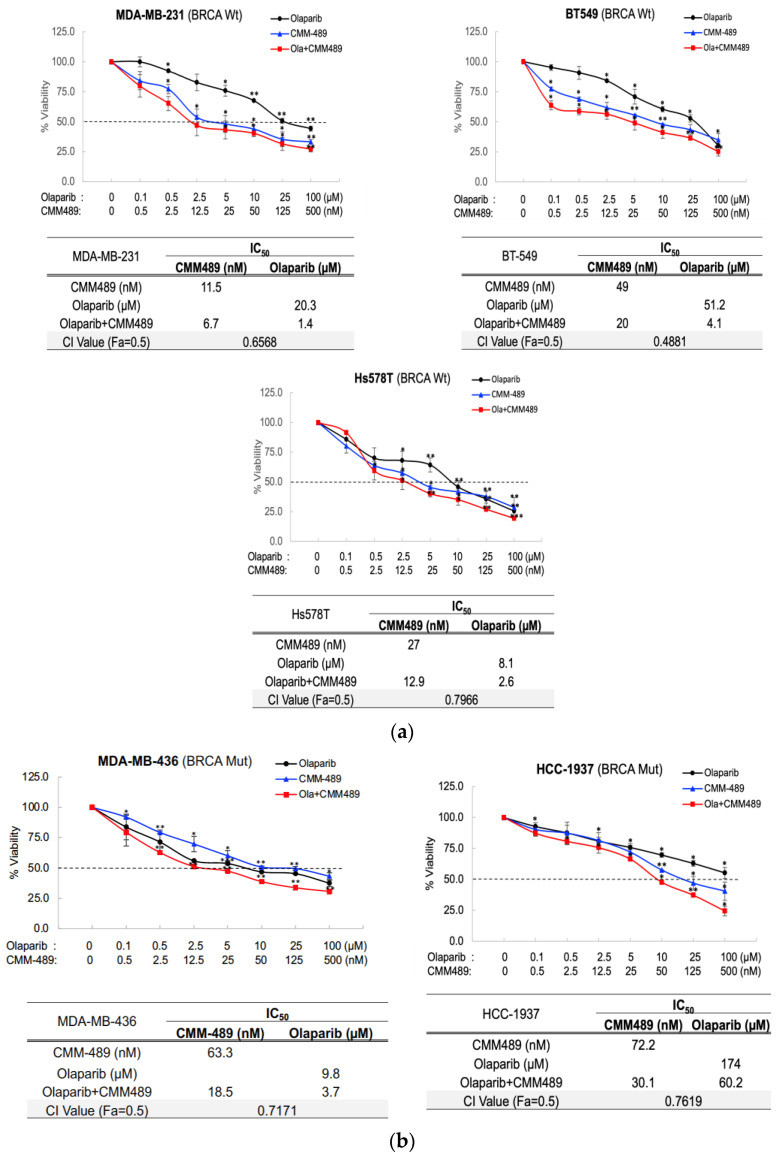
Combination index (CI) value at IC_50_ for the drug/combo of Olaparib and CMM489 in the indicated TNBC cell lines. (**a**) BRCA wild-type TNBC cells (BRCA Wt) and (**b**) BRCA mutant-type TNBC cells (BRCA Mut) were treated with the indicated concentrations of CMM489, Olaparib, or the combination (Ola + CMM489) for 3 days. Cell viability was measured using an MTT assay. The combination index (CI) in cells was calculated using CompuSyn software. The values of IC_50_ and CI under drug treatment are shown in the tables. Synergy; CI < 0.9, additive; CI = 0.9, antagonistic; CI > 0.9. All data represent the mean of at least three independent experiments ±SD. The statistical analysis was performed using Student’s *t*-test. *, *p* < 0.05, **, *p* < 0.01. (**c**) % cell viability after various doses of CMM489 treatment. All viability rates were measured with an MTT assay. Column, the mean from five independent experiments. Student’s *t*-test; * *p* < 0.05, ** *p* < 0.01, *** *p* < 0.001.

**Figure 2 cells-13-00049-f002:**
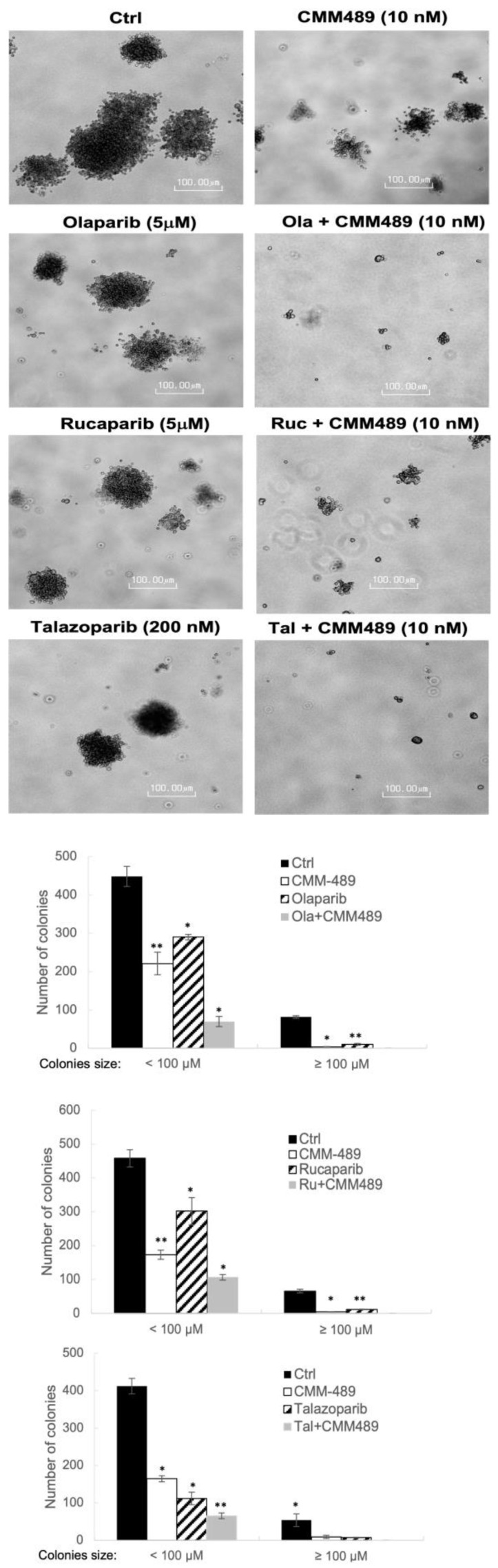
CMM489 synergized with multiple FDA-approved PARP inhibitors in the colony-forming assay. MDA-MB-231 cells were cultured in a soft-agar-containing growth medium and treated with the indicated drugs. After 4 weeks, colony formation was captured at 10 × magnification. Colonies ≥0.1 mm or <0.1 mm in diameter were counted. Representative bright-field images were acquired at least 3 times. Right panels indicate the quantitative data. Column, the mean from three independent experiments. Student’s *t*-test; * *p* < 0.05, ** *p* < 0.01.

**Figure 3 cells-13-00049-f003:**
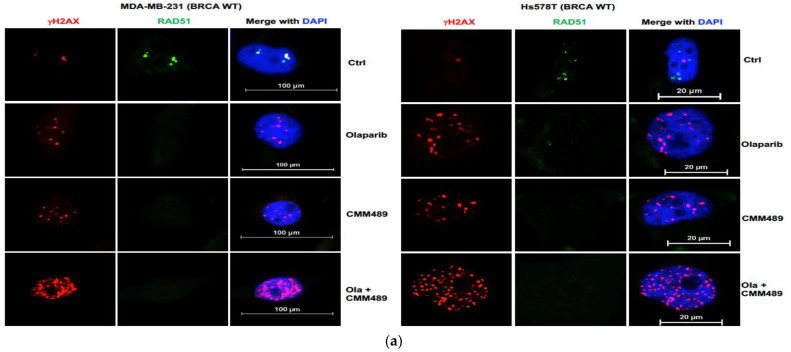
CMM489 effectively synergizes with Olaparib to increase HR deficiency. (**a**,**b**) Cells were treated with CMM489, Olaparib, or the combination (Ola + CMM489) for 3 days. RAD51 (green) and γ-H2AX (red) foci in cells were imaged via immunofluorescence microscopy. Nuclear DNA was labeled with DAPI (blue). Representative images were selected from three independent experiments. (**c**) The expression levels of RAD51 and γ-H2AX in cells treated with indicated drugs were confirmed via Western blot analysis. β-actin was used as a loading control. All blot images are representative data of three independent experiments.

**Figure 4 cells-13-00049-f004:**
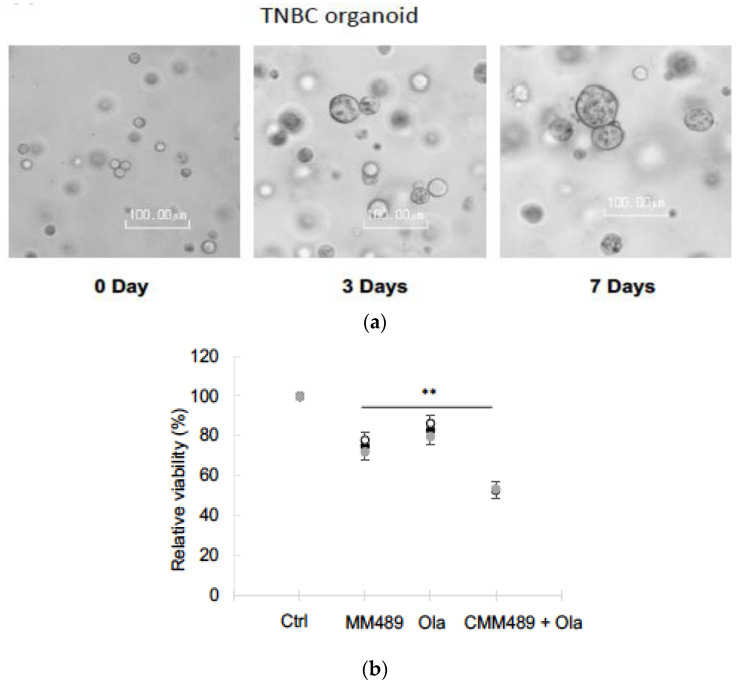
Synergistic effects of CMM489 with Olaparib were confirmed in TNBC organoid models. (**a**) Bright-field images of TNBC organoids grown in the Matrigel, demonstrating organoid outgrowth. Scale bars, 100 μm. (**b**) The dot plot represents the relative viability of TNBC organoids treated with the Ctrl (negative control), CMM489, Olaparib, and combination (CMM489 + Olaparib). N =  3. Error bars represent the SD. ** *p*  <  0.01.

**Figure 5 cells-13-00049-f005:**
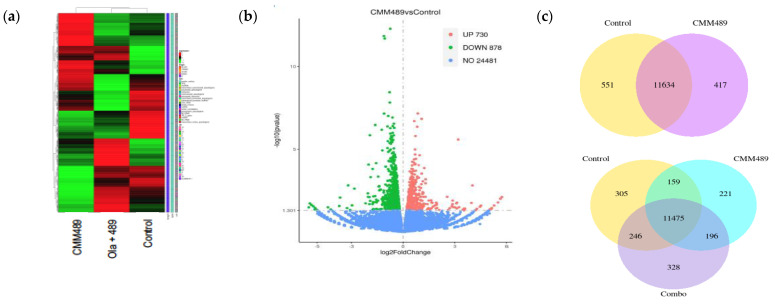
RNA-seq-identified differentially expressed genes in TNBC cells treated with CMM489 compared to the control. (**a**) The hierarchical clustering heat map of RNA sequencing represents all differentially expressed genes in the Ctrl, CMM489, and combination groups. (**b**) Volcano plots show up- (red) and downregulated genes (green) in CMM489 compared to the control. Blue dots indicate no significant difference. Screening threshold value: |log2(fold change)| > 1, corrected *p*-value < 0.05. (**c**) The Venn diagram shows overlapping differential genes in TNBC cells affected by CMM489 or Olaparib + CMM489.

**Figure 6 cells-13-00049-f006:**
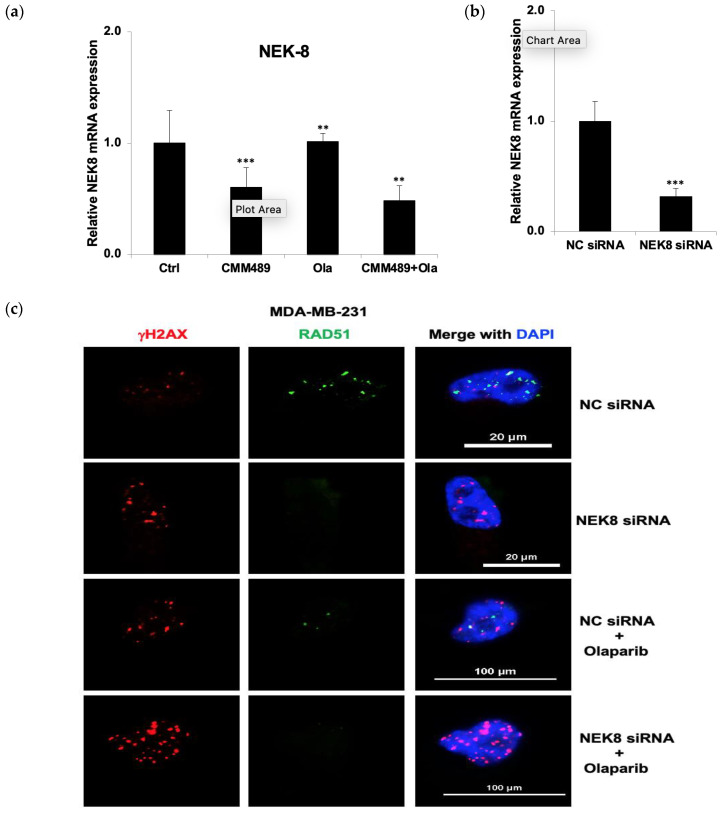
The reduction in NEK8 expression mimics the action of CMM489 in TNBC cells. (**a**) NEK8 mRNA levels relative to GAPDH in MDA-MB 231 cells treated with CMM489, Olaparib, and a combination (CMM489 + Olaparib) were measured via qRT-PCR. The graph shows the average of three independent experiments. ** *p*  <  0.01, *** *p* < 0.001. (**b**) MDA-MB-231 cells were transfected with siRNA NEK8 and the siRNA negative control (NC siRNA). The knockdown of NEK8 was confirmed via qRT-PCR. GAPDH was used as an internal control. *** *p* < 0.001. (**c**) MDA-MB-231 cells transfected with NC siRNA and NEK8 siRNA were treated with Olaparib and then subjected to immunofluorescence staining with DAPI (blue), anti-RAD51 (green), and anti-γ-H2AX (red) antibodies. Representative images were selected from three independent experiments. Bars, 20 μm or 100 μm.

**Table 1 cells-13-00049-t001:** Olaparib sensitivity and characteristics of TNBC cells used in this study.

Cell Line	Subtype	BRCA Mutation Status	Olaparib Sensitivity
MDA-MB-231	Mesenchymal Stem-like	Wild type	No
Hs578T	Mesenchymal Stem-like	Wild type	Yes
BT-549	Mesenchymal	Wild type	No
MDA-MB-436	Mesenchymal Stem-like	Mutant (5396 + 1G > A)	Yes
HCC-1937	Basal-like	Mutant (5382insC)	No

**Table 2 cells-13-00049-t002:** Top 15 genes significantly downregulated with CMM489. NEK8 is highlighted in bold and underlined.

Gene	log2 Fold Change	*p*-Value	Gene Description
TMEM63A	−1.112	1.4 × 10^−12^	transmembrane protein 63A
ADM2	−1.569	6.9 × 10^−5^	adrenomedullin 2
TNFSF18	−2.008	8.5 × 10^−5^	TNF superfamily member 18
SYT2	−1.774	1.6 × 10^−4^	synaptotagmin 2
YPEL1	−4.198	4.3 × 10^−2^	yippee-like 1
DBH	−1.888	4.9 × 10^−4^	dopamine beta-hydroxylase
** NEK8 **	** −1.389 **	** 1.7 × 10^−3^ **	** NIMA related kinase 8 **
LPAR2	−1.376	3.1 × 10^−3^	lysophosphatidic acid receptor 2
NEDD9	−1.625	6.7 × 10^−3^	neural precursor cell expressed, developmentally downregulated 9
CALCR	−1.719	7.1 × 10^−3^	calcitonin receptor
ESPN	−2.026	7.5 × 10^−3^	espin
TNS4	−1.458	1.3 × 10^−2^	tensin 4
STAB2	−1.912	1.3 × 10^−2^	stabilin 2
PIP5KL1	−1.572	1.5 × 10^−2^	phosphatidylinositol-4-phosphate 5-kinase like 1
CCR7	−2.177	1.9 × 10^−2^	C-C motif chemokine receptor 7

## Data Availability

The data presented in this study are available upon reasonable request from the corresponding author.

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
