# Peer review of "The Sensitization of Triple-Negative Breast Cancers to Poly ADP Ribose Polymerase Inhibition Independent of BRCA1/2 Mutation Status by Chemically Modified microRNA-489"

_cells, 2023, doi:10.3390/cells13010049_

Round 1
Reviewer 1 Report
Comments and Suggestions for Authors
The manuscript from Soung et al. reported 5-Fluorouridine modified miR-489, CMM489, could synergize with PARP inhibitors to treat Triple-Negative Breast Cancers (TNBC). The mechanism of this synergizing effect origins in blockade of homologous recombination (HR) in TNBC cells upon DNA damages. The work appears to be comprehensive, well-conducted, and will be well received attention by TNBC field. The further clinic study should also be promising. However, there are several issues the authors should address.
1. In Figure 1, the authors used the ratio of Olaparib to CMM-489 is 5:1, they author did not provide any rationale of this ratio, or did they observe the same result with different ratio. If they changed ratio, could they have better synergizing effect?
2. In Figure 1C, the concentration of 5-FU should be annotated in the legend or figure.
3. In Figure 3, the imaging intensities from more cells should be quantified to make convincing statement.
4. In Figure 3C, the (3C) should be on top of the western blot result.
5. The same problem happens in Figure 4,5 and 6, the panel annotation is misleading.
Comments on the Quality of English Language
Minor editing of English language required.
Author Response
We would like to thank you for the reviewer who made the constructive comments to improve our manuscript. We addressed the concerns of the reviewers as followings.
Reviewer #1:
- In Figure 1, the authors used the ratio of Olaparib to CMM-489 is 5:1, they author did not provide any rationale of this ratio, or did they observe the same result with different ratio. If they changed ratio, could they have better synergizing effect? The concentration of olaparib was in micro-molar range and the that of CMM489 was in nano-molar range. Therefore, the ratio is not 5:1. We used various doses of olaparib and CMM489 in combination and Fig. 1b shows the range that shows the maximal synergy between olaparib and CMM489.
- In Figure 1C, the concentration of 5-FU should be annotated in the legend or figure.: 5FU concentration is now added to the figure and the legend.
- In Figure 3, the imaging intensities from more cells should be quantified to make convincing statement. : The showing image represents one of multiple cells that we observed in the microscopy. If we include multiple cells in one figure, it will be difficult to provide the good resolution of fluoresced dot images.
- In Figure 3C, the (3C) should be on top of the western blot result.: According to the Journal format, (c) should be added, not (3C). As the figure legend shows Fig. 3, the readers can understand (c) represents Fig. 3C.
- The same problem happens in Figure 4,5 and 6, the panel annotation is misleading: For annotation of Figs 3-6, we followed format and instruction of Journal of Cells.
Reviewer 2 Report
Comments and Suggestions for Authors
General comments
The study described in the manuscript addresses an interesting aspect related to the need for therapeutic improvement for triple-negative breast cancer. The authors describe their results on exploiting the potential of CMM489 (Chemically Modified MiR-489) in sensitizing TNBC cells to PARP inhibitors. They found that CMM489 synergizes with multiple PARP inhibitors regardless of BRCA mutation status, improving therapeutic response. Using RNA-seq analysis and functional studies, they also identified NEK8 as a potential CMM489 target gene that mediates homologous recombination. The study is well-designed and performed and deserves to be published.
Specific comments
1.The CMM 489 agent spelling must be standardized; it appears in different ways in the text: CMM489 or CMM-489 or CMM 489 or MM489.
2.In the Material and Methods section, Nova seq 6000 should be replaced by NovaSeq 6000 system.
3.In the Material and Methods section, the authors should provide information on how the statistical analysis was performed.
4.Figure 1: (c) The treatment was with CMM489 or 5-FU, as indicated in Figure?
5.Figure 4: The cells were treated with CMM489 or MM489. Please correct. In the experiments, did control cells receive the vehicle? Which? DMSO, DMF, etc…
6.In the discussion section, the authors should highlight the limitations of their study.
7.The English has to be reviewed to improve the quality and clarity of the manuscript.
Comments on the Quality of English LanguageThe English has to be reviewed to improve the quality and clarity of the manuscript.
Author Response
We would like to thank you the reviewer who made constructive comments to improve our manuscript. We addressed the concerns of the reviewers as follows.
1. The CMM 489 agent spelling must be standardized; it appears in different ways in the text: CMM489 or CMM-489 or CMM 489 or MM489.: We used CMM489 throughout the manuscript in the revision.
2. In the Material and Methods section, Nova seq 6000 should be replaced by NovaSeq 6000 system. : We used the expression of ‘NovaSeq 6000 system’ in the material and method section.
3. In the Material and Methods section, the authors should provide information on how the statistical analysis was performed.: We provided information regarding how the statistical analysis was done in the material and method section.
4. Figure 1: (c) The treatment was with CMM489 or 5-FU, as indicated in Figure? In Fig. 1C, we demonstrated that the 5-FU component is not toxic in the concentration where CMM489 is being used. Therefore, the treatment was with 5-FU only, not CMM489. We made it clear in the result section.
5. Figure 4: The cells were treated with CMM489 or MM489. Please correct. In the experiments, did control cells receive the vehicle? Which? DMSO, DMF, etc…: MM489 was a typo (Fig. 4B) and corrected with CMM489 in the revision. CMM489 can get into the cells without a vehicle as mentioned in the introduction section (line 81). Therefore, vehicle control was not necessary.
6. In the discussion section, the authors should highlight the limitations of their study.: Despite the discovery of NEK8 as a potential mediator of CMM489 in regulating HRD pathway, we acknowledge in the discussion section that other genetic or epigenetic components downstream of CMM489 may also play important roles in HRD pathways. Identification of additional targets is the limitation of our current study and will be the focus of our future study.
7. The English has to be reviewed to improve the quality and clarity of the manuscript.: Additional English editing has been performed.
Reviewer 3 Report
Comments and Suggestions for Authors
By combining phenotypic analysis and RNA-sequencing, the authors have presented evidence suggesting that chemically modified miR-489, CMM489, may hold enhanced therapeutic potential in sensitizing DNA damage response inhibitors to suppress TNBCs. However, it is regrettable that the findings appear to lack sufficient support due to incomplete analysis of the RNA-seq data and the absence of crucial in vivo experimental evidence.
1. Does CMM489 selectively target cancer cells? Are there potential toxic effects on normal cells such as epithelial or fibroblast cells at the concentrations tested? The authors should provide experimental data to address these important questions.
2. It is essential to elucidate the in vivo mechanisms underlying the cancer suppression effect of CMM489. This information would greatly strengthen the study's conclusions.
3. Figure quality need to be improved. For instance, all scale bars should be consistently positioned in the same location within each figure presented in Figure 4A.
4. Figure 5 appears to be confusing. The details of Figure 5A are unclear. Additionally, there is a discrepancy in the total count of differentially expressed genes between Figure 5B and Figure 5C. The authors should clarify this inconsistency. Furthermore, it would be valuable to determine if the differentially expressed genes in the CMM489 and Combo groups are enriched in genes related to the DNA damage response.
Author Response
We would like to thank you for the reviewer who made the constructive comments to improve our manuscript. We addressed the concerns of the reviewers as followings.
- Does CMM489 selectively target cancer cells? Are there potential toxic effects on normal cells such as epithelial or fibroblast cells at the concentrations tested? The authors should provide experimental data to address these important questions. We addressed the toxicity issue of CMM489 in our previous study (Soung et al. Cancers 2020) by showing that there were no observable toxicities such as hair loss or weight loss in all animals that were treated with CMM489. We also demonstrated that 5-FU component of CMM489 is not toxic in the dose that we used in Fig. 1C.
- It is essential to elucidate the in vivo mechanisms underlying the cancer suppression effect of CMM489. This information would greatly strengthen the study's conclusions. Therapeutic effects and mechanism of CMM489 in vivo has been demonstrated in our previous study (Soung et al. Cancers 2020). We agree the importance of in vivo studies showing the synergy between CMM489 and PARP inhibitors, which will be the focus in our future studies. Additional in vivo assays during 10-day revision period is not feasible.
- Figure quality need to be improved. For instance, all scale bars should be consistently positioned in the same location within each figure presented in Figure 4A. Scale bars were inserted automatically when we took pictures from the microscopy. We moved the position of the pictures to push the scale bar a little bit in the similar position.
- Figure 5 appears to be confusing. The details of Figure 5A are unclear. Additionally, there is a discrepancy in the total count of differentially expressed genes between Figure 5B and Figure 5C. The authors should clarify this inconsistency. Furthermore, it would be valuable to determine if the differentially expressed genes in the CMM489 and Combo groups are enriched in genes related to the DNA damage response. Top 15 significantly downregulated genes in CMM489 represent in Table. 2. So far, NEK8 is the only gene that regulates the DNA damage response. Additional studies of differently regulated genes outside the top15 genes will be done in the near future, but not during this revision period. Fig. 5B represents Volcano plots, which is the scatter plot that shows statistical significance (P value) versus magnitude of change (fold change) whereas Fig. 5C is the Venn diagram shows the overlap of differentially regulated genes without statistical significance and fold change.
Round 2
Reviewer 3 Report
Comments and Suggestions for Authors
1. The analysis of the RNA-seq data in Figure 5 lacks depth in its interpretation. It is essential for the author to delve into a more comprehensive discussion. Specifically, there should be an exploration of the enrichment of differentially expressed genes in CMM489 within specific pathways and how these pathways influence the cellular phenotype.
2. Furthermore, the manuscript should address the synergistic effects of CMM489+Olaparib as manifested in the RNA-seq data. This should include a thorough exploration of genes that are commonly upregulated and downregulated in both CMM489 and CMM489+Olaparib conditions. A comprehensive discussion of these aspects will significantly enhance the scientific rigor and impact of the study.
3. In Figure 6a, the authors should check the raw data, as it appears that the mRNA levels of NEK-8 in both the control group and the Olaparib-treated group do not show significant differences.
Author Response
We would like to thank you for the reviewer #3 who made the constructive comments to improve our manuscript. We addressed the concerns of the reviewers as followings.
Before we mention each comments, we would like to mention our rationale to focus on target genes and pathways of CMM489 only in the current study, rather than those of olaparib or CMM489+olparib. PARP inhibitors such as olaparib prohibit dissociation of PARP enzymes from the DNA, which causes single stranded breakage of DNA. Unless this breakage is repaired efficiently, cells undergo apoptosis. Therefore, we focused on the downstream target genes and pathways of CMM489, and tried to identify the ones that may play an important role in DNA repair to explain the synergy between CMM489 and olaparib. In other words, the synergy likely come from DNA breakage from PARP inhibitors, and the blockage of DNA repair response from CMM489, rather than combination of differential regulation of genes from olaparib. Olaparib mediated single stranded breakage of DNA will lead to double-stranded breakage unless it is repaired, which will lead to massive cell stress responses and random apoptotic responses. Therefore, the focus of our current studies is to identify CMM489 downstream target gene or pathway that block DNA repair of the breakage created by olaparib. The initial round of pathway analysis does not show any major DNA repair pathway from RNA seq analysis. Therefore, we analyzed each of top differentially regulated genes by CMM489 treatment and identified NEK8 is the major player that explains the synergy effects from CMM489. We added this description in the discussion section (highlighted in blue color) to avoid the confusion.
- As mentioned above, the original analysis of signaling and functional pathways regulated by CMM489 did not show DNA damaging responses and homologous recombination. That’s why we analyzed each of the top target genes regulated by CMM489 and led to the discovery of NEK8 as a major target gene to explain synergy between CMM489 and olaparib. We added this description in the discussion section.
- As also mentioned above, the massive apoptotic events will occur under CMM489 and olaparib co-treatment conditions, and gene analysis likely to show random up regulation of cell stress response genes prior to apoptosis. We added the description of the rationale why we target downstream of CMM489 only in the discussion section.
- NEK8 is the downstream target of CMM489 and should be only regulated by CMM489, but not by olaparib. Olaparib induces the single stranded breakage of DNA, which requires a DNA repair process. Otherwise, cancer cells will undergo apoptosis. CMM489 block DNA damaging responses via NEK8. These two are separate events, which is the basis of “synthetic lethality”. In other words, the synergy does not come from collaborative down-regulation of NEK8 by CMM489 and olaparib. Olaparib causes DNA damage, and CMM489 blocks DNA repair by down-regulating NEK8.
Round 3
Reviewer 3 Report
Comments and Suggestions for Authors
In Figure 6a, the authors should review the raw data because the mRNA levels of NEK-8 in the olaparib-treated group do not appear to exhibit significant differences compared to the control group.